# Mechanisms of Immune Evasion in HIV-1: The Role of Virus-Host Protein Interactions

**DOI:** 10.3390/cimb47050367

**Published:** 2025-05-16

**Authors:** Antonios Mouzakis, Vasileios Petrakis, Eleni Tryfonopoulou, Maria Panopoulou, Periklis Panagopoulos, Katerina Chlichlia

**Affiliations:** 1Laboratory of Molecular Immunology, Department of Molecular Biology and Genetics, Democritus University of Thrace, 68100 Alexandroupolis, Greece; antonios.n.mouzakis@gmail.com (A.M.); etryfono@mbg.duth.gr (E.T.); 2Second Department of Internal Medicine, University General Hospital Alexandroupolis, Democritus University of Thrace, 68100 Alexandroupolis, Greece; vasilispetrakis1994@gmail.com (V.P.); ppanago@med.duth.gr (P.P.); 3Laboratory of Microbiology, University General Hospital Alexandroupolis, Democritus University of Thrace, 68100 Alexandroupolis, Greece; mpanopou@med.duth.gr

**Keywords:** HIV regulatory proteins, immune evasion, host-cell proteins, interactions

## Abstract

This review explores the mechanisms by which Human Immunodeficiency Virus type 1 (HIV-1) regulatory proteins manipulate host cellular pathways to promote viral replication and immune evasion. Key viral proteins, such as Nef, Vpu, Vif, Vpr, and Env, disrupt immune defenses by downregulating surface molecules such as CD4 (Cluster of Differentiation 4) and Major Histocompatibility Complex (MHC) class I, degrading antiviral enzymes like APOBEC3G (Apolipoprotein B mRNA editing catalytic polypeptide-3G) and SAMHD1 (Sterile Alpha Motif and Histidine Aspartate domain-containing protein 1), and counteracting restriction factors including BST-2 (Bone Marrow Stromal Antigen 2)/Tetherin and SERINC5 (Serin Incorporator 5). These interactions support viral persistence and contribute to the establishment of chronic infection. Emerging therapeutic strategies aim to disrupt these HIV-host interactions to restore innate antiviral responses and enhance immune clearance. Approaches such as stabilizing host restriction factors or blocking viral antagonists offer a promising alternative to conventional antiretroviral therapy. By targeting host-dependent pathways, these interventions may reduce drug resistance, tackle latent reservoirs, and provide a pathway toward sustained viral remission or functional cure.

## 1. Introduction

Human Immunodeficiency Virus (HIV), first identified in the early 1980s, is a retrovirus that targets and weakens the immune system by infecting CD4+ T cells. If left untreated, HIV can progress to Acquired Immunodeficiency Syndrome (AIDS). As of 2023, more than 38 million people worldwide are living with HIV [1]. The virus is primarily transmitted through blood, sexual contact, and from mother to child during childbirth or breastfeeding [2]. Two types of HIV have been identified: HIV-1, the most prevalent and aggressive form, and HIV-2, which is less common and largely confined to West Africa [3].

Once inside the body, HIV integrates into the host genome, establishing a chronic infection that gradually impairs immune function. This decline increases susceptibility to opportunistic infections and cancers [4]. The virus’s high mutation rate and sophisticated immune evasion strategies have made the development of a vaccine or a definitive cure particularly challenging [5,6]. Although antiretroviral therapy (ART) has proven effective in suppressing viral replication and restoring immune function, allowing many individuals to manage HIV as a chronic condition, significant challenges remain. These include the emergence of drug resistance, issues with long-term treatment adherence, disparities in access to healthcare, and persistent mechanisms of viral immune escape [7].

Immune evasion is a hallmark of many pathogens, including viruses, bacteria, protozoa, and helminths. Immune evasion strategies, ranging from avoiding immune recognition to actively suppressing host immune responses, disrupt immune homeostasis and complicate pathogen clearance [8,9,10]. HIV-1, in particular, has evolved sophisticated mechanisms to persist in the host, including high mutation rate, downregulation of immune markers, infection of key immune cells, suppression of immune signaling, and the establishment of latent viral reservoirs. While ART has significantly improved viral suppression and immune preservation, these evasion strategies remain major obstacles to viral eradication [9,10]. A deeper understanding of the molecular interactions between HIV and host immune pathways is essential for advancing therapeutic strategies. Recent efforts have focused on targeting host-virus interactions through latency-reversing agents, broadly neutralizing antibodies, and innate immune modulators. This review outlines the current knowledge of HIV–host protein interactions that facilitate immune evasion, with an emphasis on their therapeutic implications.

### 1.1. HIV Virion and Genome

HIV-1 is a retrovirus with a complex structure (Figure 1a) that facilitates infection. Its envelope, derived from the host membrane, contains the glycoproteins gp120 and gp41. Gp120 binds to CD4 and co-receptors such as C-C Chemokine Receptor Type 5 (CCR5) or C-X-C Chemokine Receptor Type 4 (CXCR4), while gp41 facilitates fusion with the host membrane. Beneath the envelope is the matrix protein p17, which is crucial for structural integrity and viral assembly. The capsid, composed of p24, protects the RNA and associated enzymes, while the nucleocapsid protein p7 stabilizes the RNA [11,12]. The HIV-1 genome consists of two single-stranded RNA molecules that reverse-transcribe into DNA upon entering the host cell. Essential enzymes, including reverse transcriptase, integrase, and protease, are required for replication and maturation of new virions [3].

The HIV-1 genome (Figure 1b), approximately 9700 to 10,000 nucleotides in length, encodes proteins and enzymes necessary for replication and interaction with the host [3,11,12]. Structural genes (*gag*, *pol*, *env*) code for viral proteins, while regulatory genes (*tat*, *rev*, *nef*, etc.) modulate replication and immune evasion. Untranslated regions (UTRs) and long terminal repeats (LTRs) regulate replication and packaging. This genomic architecture enables HIV-1 to replicate efficiently, evade the immune system, and persist in the host [8,9].

### 1.2. HIV-1 Life Cycle

The HIV-1 life cycle (Figure 1c) begins when the virus enters a CD4+ T cell. The glycoprotein gp120 binds to the CD4 receptor, inducing a conformational change that enables interaction with the co-receptors CCR5 or CXCR4. This interaction facilitates the fusion of the viral envelope with the host cell membrane, delivering the viral core into the cytoplasm. Reverse transcription then occurs, during which the viral RNA is converted into double-stranded DNA by reverse transcriptase. This newly synthesized viral DNA is integrated into the host genome by the viral integrase, forming a provirus. Subsequently, the host cell’s transcriptional and translational machinery is hijacked to produce viral mRNA and synthesize viral proteins as well as new RNA genomes. During assembly, viral proteins and RNA are packaged into a nucleocapsid core. The Gag polyprotein is cleaved into functional components, such as p17, p24, and p7, essential for virion structure and function. These components assemble at the host cell membrane to form new virions. Budding follows, with nascent virions acquiring their envelope from the host membrane. Viral protease then cleaves polyproteins into their mature forms, essential for infectivity. The mature virions are released and can infect other CD4+ cells, continuing the infection cycle [11,12].

### 1.3. HIV-1 Reservoir Cells

The establishment of the latent HIV-1 reservoir occurs early in infection, often within days of viral exposure. Upon entry into the body, HIV-1 primarily targets activated CD4+ T cells, integrating its genetic material into the host genome. While many of these infected cells produce new virions and subsequently die, a subset transitions into a resting state, in which the virus remains transcriptionally silent (dormant). These latently infected cells, particularly memory CD4+ T cells, can persist in the body for years without producing viral proteins, allowing them to evade immune surveillance and ART. Moreover, HIV-1 may establish latency in other cell types, including macrophages and dendritic cells (DCs), further complicating eradication efforts. In T cells, the stability of the latent reservoir is reinforced by homeostatic proliferation, wherein latently infected cells divide without reactivating the virus, thereby ensuring its long-term persistence [13].

Reservoir cells play a critical role in the persistence of HIV-1 infection despite effective ART. These cells, primarily comprised of latently infected CD4+ memory T cells, harbor integrated, replication-competent proviral DNA. Even after prolonged ART, which effectively suppresses active viral replication, the latent reservoir remains a significant barrier to cure. Upon interruption of ART, these latently infected cells can become reactivated, resulting in viral rebound. Understanding the mechanisms underlying reservoir persistence is essential for developing strategies to eliminate or reduce the HIV-1 reservoir [14].

### 1.4. Immune Evasion

HIV-1 employs several strategies to evade immune detection, one of the most prominent being its exceptionally high mutation rate. This is primarily due to the error-prone reverse transcriptase enzyme, which lacks proofreading capability during the conversion of viral RNA into DNA. As a result, frequent mutations occur during replication, allowing the virus to continuously alter its epitopes and escape immune recognition. Another key mechanism involves the downregulation of Major Histocompatibility Complex (MHC) class I and II molecules on infected cells. This impairs antigen presentation and helps the virus avoid detection by cytotoxic T lymphocytes (CTLs) and other immune effectors [15,16].

HIV-1 also targets key immune cells, particularly CD4+ T cells, which are essential for orchestrating immune responses. By depleting these cells, HIV compromises the immune system, especially the adaptive immune response. Furthermore, the virus can evade neutralizing antibodies due to its high mutation rate, which enables rapid alterations in the structure of envelope proteins such as gp120 and gp41. These mutations reduce the effectiveness of antibodies by impairing their ability to recognize and neutralize the virus [15,17].

HIV-1′s tropism for CD4+ T cells, and its dependence on specific co-receptors further enhance its immune evasion capabilities. In the early stages of infection, the virus predominantly uses a CCR5 co-receptor (R5-tropic virus), but in later stages, it may switch to CXCR4 (X4-tropic virus). This co-receptor switching contributes to immune escape and disease progression [15,17].

Moreover, HIV-1 produces immunosuppressive proteins that contribute to immune system dysfunction. For example, the viral protein Tat interferes with the function of various immune cells, including CD4+ T cells and dendritic cells. The virus also establishes latent reservoirs in various tissues, such as the gut-associated lymphoid tissue, lymph nodes, and the central nervous system. These reservoirs protect the virus from immune clearance, as the virus is not actively replicating, and the infected cells do not present viral antigens. The persistence of these reservoirs contributes to chronic immune activation and systemic inflammation, further weakening the immune system [18].

Over time, chronic HIV-1 infection leads to the progressive depletion of CD4+ T cells, especially in untreated individuals. This gradual loss of helper T cells compromises immune competence, rendering the host increasingly susceptible to opportunistic infections and malignancies. In addition to directly infecting and destroying CD4+ T cells, HIV-1 disrupts the broader immune response by impairing the function of other immune cells, including macrophages and natural killer (NK) cells. These multifaceted strategies facilitate viral persistence and replication, posing substantial barriers to eradication despite significant advancements in antiretroviral therapies [19].

Notably, HIV-host interactions play a central role in immune evasion and the establishment of long-term infection. Through an array of sophisticated mechanisms, HIV-1 modulates host immune pathways to enable persistent replication despite active immune surveillance. The virus encodes several proteins that downregulate key immune molecules and inhibit antiviral defense mechanisms. These interactions between HIV-1 and host immune components not only support viral propagation but also contribute to immune escape, thus sustaining chronic infection. Elucidating these interactions is essential for understanding how HIV-1 circumvents host defenses and for guiding the development of targeted therapeutic strategies.

## 2. HIV Virus-Host Cell Proteins Interactions

### 2.1. Host Proteins That Interact with Viral Vpu

Viral Protein U (Vpu) is an 81-amino acid, membrane-embedded phosphoprotein comprising three α-helices: the N-terminal transmembrane domain (Helix 1) and two C-terminal helices (Helices 2 and 3). Helix 2 is amphipathic and partly embedded in the membrane, while Helix 3 contains the serine residues S52 and S56, which are phosphorylated by casein kinase 2 (CK-2). This phosphorylation enables interaction with β-TrCP (BTRC/Beta-Transducin Repeat Containing), a component of the SCF (Skp1-Cullins-F-box protein) E3 ubiquitin ligase complex [20,21]. Vpu performs two primary functions in HIV-1 pathogenesis: it promotes the degradation of newly synthesized CD4 in the endoplasmic reticulum (ER) via the ubiquitin-proteasome pathway and facilitates virion release by antagonizing the host restriction factor BST-2 (Bone Marrow Stromal Antigen 2)/Tetherin, as summarized by Khan et al. (2021) [20]. Beyond these roles, Vpu modulates various host cellular processes. It alters intracellular protein trafficking and impairs MHC class II antigen presentation [22,23], stabilizes the tumor suppressor protein p53 [24], inhibits NK cell granule release by targeting NTB-A (NK-T-B antigen), and downregulates CD1d, thereby impairing lipid antigen presentation to Natural Killer T (NKT) cells [20]. Vpu also induces apoptosis [24] and interferes with CD4+ T-cell migration by downregulating the chemokine receptor CCR7 [25].

CD4, the primary receptor for HIV-1 entry, is downregulated post-infection to prevent superinfection, enhance viral replication, and evade immune surveillance. Sustained CD4 expression can hinder viral replication by binding to Env precursors (gp160) in the ER, thereby blocking their proper maturation into gp41 and gp120. Vpu mediates CD4 degradation by retaining it in the ER through interactions involving their respective transmembrane domains. This retention leads to polyubiquitination and subsequent ER-associated degradation (ERAD), a process that requires the recruitment of BTRC and the SCF E3 ubiquitin ligase complex [20].

BST-2 is a 30–36 kDa type II transmembrane protein upregulated by type I interferons, which restricts the release of HIV-1 by physically tethering budding virions to the host cell membrane. This restriction is facilitated through the formation of disulfide-linked homodimers and the presence of glycosylphosphatidylinositol (GPI) anchors. HIV-1 Vpu counteracts BST-2 through three primary mechanisms: (1) downregulation of BST-2 from the cell surface via clathrin/AP2 (Adaptor Protein Complex 2)-mediated endocytosis, (2) inhibition of BST-2 recycling and its transport to the plasma membrane, and (3) SCF-BTRC-mediated ubiquitination followed by lysosomal degradation, targeting lysine, serine, and threonine residues. Unlike CD4, which is directed to the proteasome, BST-2 is predominantly degraded in the lysosome, a process requiring the Endosomal Sorting Complex Required for Transport (ESCRT) complex and Rab7-dependent trafficking [20].

AP1M1 is a subunit of the adaptor protein complex 1 (AP-1), a clathrin-associated complex involved in sorting cargo protein between the trans-Golgi network (TGN) and endosomal compartments. AP complexes facilitate clathrin recruitment to membranes and recognize cytoplasmic sorting signals on transmembrane cargo proteins [26]. Vpu exploits AP1M1 to interfere with host trafficking pathways, thereby promoting the degradation of BST-2 and enhancing viral egress. This interaction represents a critical mechanism by which HIV-1 subverts host cellular machinery to its advantage [27].

Moreover, HIV-1 infection leads to a marked reduction in endogenous levels of P-selectin glycoprotein ligand-1 (PSGL-1) in CD4+ T cells. This effect is not observed in HIV strains lacking Vpu, underscoring the critical role of Vpu in its regulation. Direct interaction between Vpu and PSGL-1 was confirmed through co-immunoprecipitation assays. Given Vpu’s established role in hijacking the SCF-BTRC E3 ligase complex, researchers investigated whether PSGL-1 is similarly targeted for degradation via this pathway. Experimental results revealed that Vpu induces the ubiquitination and degradation of PSGL-1 in a SCF-BTRC-dependent manner. Notably, siRNA-mediated knockdown of BTRC abolished this effect, further confirming the involvement of this pathway. In addition, Vpu mutants lacking functional serine residues at positions 52 and 56 (S52/56N or S52N/S56N), which are required for BTCR binding, failed to induce PSGL-1 ubiquitination and degradation. Collectively, these findings support a model in which PSGL-1 acts as an effective host restriction factor antagonized by HIV-1 Vpu via SCF-BTRC-mediated degradation, thereby contributing to immune evasion and viral persistence [28,29].

Guanylate-Binding Protein-5 (GBP5) belongs to the superfamily of the interferon-inducible guanosine triphosphatase (GTPase) and plays a critical role in the innate immune response against various pathogens, including bacteria, protozoa, and viruses. GBP family members exert their function by hydrolyzing guanosine triphosphate (GTP) to produce guanosine diphosphate (GDP) and guanosine monophosphate (GMP). GBP5 is predominantly localized in the cytosol and on endosomal membranes. However, during HIV-1 infection, it has been observed that it co-localizes with HIV-1 particles and significantly diminishes the production of infectious virions [28].

Data from the Genomic Utility for Association and Viral Analysis in HIV (GuavaH) database indicate that GBP5 expression is upregulated in HIV-1-infected individuals [28]. Furthermore, evidence suggests that the HIV-1 Tat protein may stimulate GBP5 expression in primary human T cells [30]. GBP5 exerts its antiviral activity by interfering with the processing of HIV-1 Env within the Golgi apparatus. Specifically, GBP5 disrupts N-linked oligosaccharide glycosylation of Env, a modification essential for its maturation. This interference leads to the incorporation of immature gp160 into nascent virions, impairing Env cell surface expression and thereby reducing the virion infectivity [28].

HIV-1 has evolved mechanisms to evade GBP5-mediated restriction, notably through mutations in the start codon of the Vpu gene. Since both Vpu and Env are translated from a single bicistronic mRNA, these mutations enhance Env translation, effectively counteracting GBP5 inhibition. However, this adaptation comes at the cost of increased susceptibility to BST-2. Loss of functional Vpu compromises the virus’s ability to antagonize the host restriction factor BST-2, resulting in virion retention at the cell surface and reduced viral dissemination [28,31].

A summary of these interactions is presented in Table 1.

### 2.2. Host Proteins That Interact with Viral Nef

Negative regulatory factor (Nef) is a small, nonenzymatic accessory protein encoded by HIV-1, with a molecular weight ranging from approximately 27 to 34 kDa, depending on the specific lentiviral strain. It is expressed early during infection and maintained throughout the entire viral life cycle. Initially characterized as a negative regulatory factor that inhibited viral replication, Nef has since been recognized as a critical facilitator of HIV-1 replication, viral dissemination, and immune evasion. Structurally, Nef comprises two principal domains: an N-terminal anchor region and a folded core domain [32].

The N-terminal region, consisting of approximately 60 amino acids, is predominantly unstructured and includes a N-terminal glycine myristoylation signal. This modification mediates membrane association, anchoring the protein to intracellular membranes. The adjacent folded domain spans approximately 105 residues and contains a flexible internal loop. This structural flexibility allows the folded core domain to disengage from the membrane, thereby enabling interactions with various host cell proteins, many of which are membrane-associated. Although Nef lacks intrinsic enzymatic activity, it mediates a wide array of protein-protein interactions that contribute significantly to HIV pathogenesis, particularly by enhancing viral replication and facilitating immune evasion [32,33].

While CD4 functions as the primary receptor for HIV-1 entry, its persistent expression on the surface of infected cells poses several risks, including the potential for cytotoxic superinfection and the initiation of antibody-dependent cell-mediated cytotoxicity (ADCC) via CD4-Env interactions. To mitigate these effects, Nef promotes the internalization and degradation of CD4 from the plasma membrane, thereby enhancing viral fitness and enabling immune evasion [32].

Under normal physiological conditions, CD4 downregulation occurs via clathrin-mediated endocytosis, a process initiated by the interaction between the CD4 cytosolic tail and the AP-2 complex. This internalization pathway is phosphorylation-dependent and results in the trafficking of CD4 to endolysosomes for degradation. Nef enhances this pathway by directly interacting with both the cytosolic tail of CD4 and the α and σ2 subunits of AP-2 (AP2A1/2 and AP2S1), thereby accelerating the removal of CD4 from the cell surface [34,35].

Interestingly, in addition to Nef, the Vpu protein also mediates CD4 down-regulation, albeit a distinct mechanism that operates later in the viral replication cycle. Unlike Nef, Vpu targets CD4 for proteasomal degradation via a ubiquitin-dependent pathway independent of AP-2/clathrin-mediated endocytosis [32,36]. The coordinated action of Nef and Vpu ensures robust and temporally regulated depletion of surface CD4, which supports efficient viral replication and facilitates evasion of host immune responses [32].

Beyond its role in enhancing CD4 endocytosis, Nef also recruits the host protein ALG-2-interacting protein X (ALIX) to facilitate CD4 trafficking to the multivesicular body (MVB) pathway, leading to its lysosomal degradation. Nef interacts with both the Bro1 and V domains of ALIX within endosomal compartments that are enriched with internalized CD4 and MVB-associated proteins. Overexpression of the ALIX V domain competitively inhibits endogenous Nef-ALIX binding, disrupting CD4 lysosomal degradation. Similarly, depletion of ALIX impairs CD4 trafficking to lysosomes, underscoring the importance of the Nef-ALIX interaction in this process [37].

ALIX serves as a functional bridge between Nef and the ESCRT machinery, mediating the incorporation of CD4 into intraluminal vesicles (ILVs) within MVBs [37]. ALIX interacts with several ESCRT components, including TSG101 (Tumor Susceptibility Gene 101) of ESCRT-I [38] and CHMP4 (Charged Multivesicular Body Protein 1) of ESCRT-III [39], which are essential for cargo sorting and vesicle formation. Nef’s association with ALIX occurs through distinct regions of the Bro1 domain, enabling simultaneous binding of CHMP4 and Nef without mutual exclusion, thereby ensuring efficient recruitment of ESCRT components to promote CD4 degradation [37].

Nef targets CD4 for degradation via the MVB pathway, circumventing the conventional requirement for ubiquitination. Nef interacts with ALIX within endosomal compartments marked by MVB-associated proteins such as CD63, supporting the hypothesis that ALIX acts as an adaptor connecting Nef-associated CD4 to the ESCRT machinery. This process facilitates the lysosomal degradation of CD4 [37,40].

Nef’s ability to exploit the MVB/lysosomal trafficking route extends beyond CD4. Evidence suggests that Nef may similarly downregulate a variety of immune-related surface molecules, including MHC class I, CD8, CD28, CXCR4, CCR5, and the co-stimulatory proteins CD80 and CD86. These processes may also involve ALIX and components of the ESCRT machinery, although further research is required to clarify whether these targets are trafficked via the same ESCRT-dependent mechanism [37,41,42].

A particularly well-characterized target of Nef is the MHC class I antigen presentation pathway, which is crucial in presenting viral peptides on the cell surface to be recognized by CD8 cytotoxic T cells. Unlike its interaction with CD4, Nef downregulates MHC class I through mechanisms involving the AP-1 complex [32,43,44]. Two temporally distinct mechanisms have been proposed to explain this effect [32].

In the early phase, often described as the “signaling mode”, Nef is recruited to the TGN by the PACS-2 (Phosphofurin Acidic Cluster Sorting Protein 2) adaptor protein [32,45]. At this site, Nef activates Src-family kinases through with their SH3 (SRC Homology 3) domain interactions—specifically engaging Hematopoietic Cell Kinase (Hck) in macrophages and Lck/Yes novel tyrosine kinase (Lyn) in T cells—thereby initiating a signaling cascade that enhances the production of phosphatidylinositol (3,4,5)-trisphosphate via phosphoinositide 3-kinase (PI3K), which in turn activates the small GTPases Arf1 (ADP-Ribosylation Factor 1) and Arf6. These GTPases facilitate the internalization of MHC class I molecules from the cell surface. Once internalized, MHC class I forms a complex with Nef and AP-1 and is sequestered in endosomal vesicles, preventing its recycling back to the cell surface [32,44,45,46,47].

In the second mechanism, termed the “stoichiometric mode”, Nef forms a complex with AP-1 and Arf1 at the TGN to intercept newly synthesized MHC class I molecules, thereby preventing their anterograde transport to the plasma membrane. As a result, MHC class I cell surface expression is diminished, impairing the immune system’s capacity to recognize and eliminate infected cells [32,44].

Together, both mechanisms enable HIV-infected cells to evade CD8 T cell-mediated immune surveillance by reducing the presentation of viral antigens on the surface. This immune evasion significantly contributes to viral persistence and pathogenesis [32].

In addition to these pathways, evidence suggests that Nef also interacts with the adaptor proteins PACS-1 and PACS-2 in endosomal compartments to further mediate MHC class I downregulation. Although the relevant binding domains have been identified, the precise molecular mechanism underlying this process remains unclear [32].

Lck (Lymphocyte-specific protein tyrosine kinase), a Src family tyrosine kinase, plays a central role in the T cell receptor (TCR) signaling and T cell activation. In T cell lines, Nef expression has been associated with the intracellular redistribution of Lck. However, it is not yet fully understood whether this effect is due to a direct physical interaction [32,48]. Studies using ectopic co-expression of Nef and Lck in defined model systems have not shown significant alterations in Lck kinase activity, suggesting that any modulation of Lck by Nef may occur indirectly rather than through direct binding [49].

HIV-1 infection is also known to interfere with cholesterol metabolism by downregulating the cholesterol transporter ABCA1 (ATP-binding cassette transporter A1), a critical regulator of cellular cholesterol efflux. This disruption is primarily mediated by Nef, which impairs the interaction between ABCA1 and the endoplasmic reticulum chaperone calnexin, thereby inhibiting proper ABCA1 maturation and promoting its subsequent degradation [50,51]. Notably, this effect is not limited to intracellular pathways; extracellular vesicles containing Nef have also been shown to downregulate ABCA1, implying that Nef may directly interact with ABCA1 at the plasma membrane as well [52].

Nef disrupts host cell actin dynamics, thereby impairing T lymphocyte responses to chemokine signals and TCR activation. This process involves the formation of a transient multiprotein complex, in which Nef hijacks the host serine/threonine kinase PAK2 (p21-activated kinase 2) to phosphorylate and inactivate the actin-severing protein cofilin [53,54,55]. The phosphorylation-induced inactivation of cofilin prevents actin filament disassembly, resulting in the stabilization of the actin cytoskeleton within infected cells. Recent studies have identified components of the exocyst complex (EXOC), a protein complex that is involved in vesicular trafficking and actin regulation, as additional interaction partners of Nef. Notably, Nef engages EXOC through the same molecular region used for PAK2 binding [53].

Further analysis revealed that Nef-EXOC interaction occurs in HIV-infected human T lymphocytes and is evolutionarily conserved among lentiviral Nef proteins. Interestingly, this association is mediated by the PAK2 binding site on Nef and requires the presence of PAK2, specifically the Rac1/Cdc42 interaction domain, although not its kinase activity [53,56]. While EXOC was found to be dispensable for the canonical effector functions of PAK2, it was essential for Nef’s ability to inhibit actin cytoskeletal remodeling and block proximal TCR signaling. These findings suggest that Nef exploits PAK2 in a dual mechanism, utilizing both PAK2’s kinase activity and its role as an adaptor protein to recruit EXOC. This coordinated mechanism allows Nef to suppress host cell actin dynamics and effectively modulate T-cell activation [53].

In addition to manipulating host cytoskeletal responses, Nef also counteracts the antiviral function of serine incorporator 5 (SERINC5), a multipass transmembrane protein expressed on the surface of HIV-1 producer cells. When incorporated into newly assembled virions, SERINC5 interferes with viral entry by impairing membrane fusion and viral core delivery in a manner dependent on the viral Env protein [32].

Nef antagonizes this restriction factor by downregulating SERINC5 from the plasma membrane through an AP-2-dependent endocytic pathway. Following internalization, SERINC5 is directed to the endolysosomal pathway for degradation. This mechanism underscores the critical role of Nef in enhancing HIV-1 infectivity by neutralizing intrinsic host cell defenses [32].

PSGL-1 is a 120-kDa transmembrane glycoprotein expressed on the surface of various immune cells, including both myeloid and lymphoid cells. It plays a critical role in restricting HIV-1 infectivity by preventing the attachment of virions to target cells. Inflammatory stimuli elevate PSGL-1 expression, thereby facilitating leukocyte attachment and promoting transmigration into inflamed tissues. During T-cell differentiation, cytokines such as IFN-γ (Interferon-γ) and IL-12 (Interleukin-12) enhance PSGL-1 expression, indicating a potential regulatory role for IFN-γ in its induction [54]. Experimental findings demonstrate that while HIV-1 infection reduces PSGL-1 protein levels, its mRNA expression remains unchanged. Despite this post-transcriptional downregulation, PSGL-1 continues to effectively inhibit virion adhesion to cells, thereby reducing viral infectivity [28].

Nef has been implicated in the downregulation of surface PSGL-1. This reduction occurs in a concentration-dependent manner, allowing HIV-1 to partially evade the antiviral effects imposed by PSGL-1. Importantly, Nef does not significantly affect intracellular PSGL-1 levels. Instead, it appears to redistribute the protein to intracellular compartments. These findings suggest that Nef facilitates viral escape by modulating the subcellular localization of PSGL-1 rather than by promoting its degradation [28,57].

In addition to PSGL-1, interferon-induced transmembrane (IFITM) proteins play a key role in host defense against a broad range of viruses, including dengue virus, influenza A H1N1, West Nile virus, and HIV-1 [28,58,59]. IFITM proteins, which are predominantly membrane-associated, inhibit viral entry by disrupting membrane fusion events [60]. In the context of HIV-1, their antiviral activity is co-receptor-specific: IFITM1 predominantly inhibits CCR5-tropic HIV-1 strains, whereas IFITM2 and IFITM3 are more effective against CXCR4-tropic strains [61]. Beyond entry inhibition, IFITM proteins also attenuate HIV-1 replication at post-entry stages. For example, the knockdown of IFITM1 results in elevated viral titers, and IFITM expression has been associated with reduced levels of the HIV-1 Gag protein, suggesting an additional role in limiting viral protein synthesis [28,58].

Despite these antiviral effects, Nef appears to counteract IFITM-mediated restriction. Studies have shown that the expression of Nef enhances HIV-1 replication by approximately four (4)-fold in the presence of IFITM1 and IFITM2. In contrast, viruses lacking Nef exhibit significantly diminished Gag protein levels when co-expressed with IFITMs, highlighting Nef’s role in overcoming this restriction. Interestingly, this antagonism does not involve the degradation of IFITM proteins, as their overall levels remain unaffected. One proposed mechanism involves the subcellular re-trafficking of IFITM proteins, possibly diverting them away from sites where they exert their inhibitory effects on viral replication [28,58]. A recent proteomic analysis study on extracellular vesicles (EVs) derived from HIV-infected T cells has further expanded our understanding of Nef’s function. The study revealed that Nef expression significantly alters the protein composition of EVs, including a marked depletion of IFITM1, IFITM2, and IFITM. This reduction suggests that Nef may modulate IFITM trafficking, limiting their incorporation into EVs and potentially impairing their antiviral signaling roles. These findings underscore a novel mechanism through which Nef neutralizes host restriction factors, thereby facilitating efficient HIV-1 replication [62].

A summary of these interactions is presented in Table 2.

### 2.3. Host Proteins That Interact with Viral Vif

The Virion Infectivity Factor (Vif) protein of HIV-1 is a highly basic 23 kDa protein composed of 192 amino acids. Vif plays an essential role in viral replication in vivo, as deletion of the *vif* gene significantly impairs HIV-1 replication in SCID-hu (Severe Combined Immunodeficiency Humanized) mouse models and simian immunodeficiency virus (SIV) replication in macaques [63]. The primary function of Vif is to antagonize APOBEC3G (Apolipoprotein B mRNA Editing Catalytic Polypeptide-like 3G), a host cytidine deaminase that introduces G-to-A hypermutations in viral DNA during reverse transcription, potentially rendering the virus non-infectious and thus restricting HIV-1 replication [64,65].

Research has demonstrated that Vif prevents the incorporation of APOBEC3G into newly formed virions by targeting it for proteasomal degradation [66]. In addition to this degradation-dependent mechanism, some studies suggest that Vif can directly block APOBEC3G incorporation into virions through degradation-independent pathways. Moreover, Vif has been reported to inhibit APOBEC3G translation, further limiting its availability and reinforcing the suppression of its antiviral function [67]. These findings indicate that Vif employs multiple complementary mechanisms to neutralize APOBEC3G-mediated restriction.

Vif achieves APOBEC3G degradation by assembling an E3 ubiquitin ligase complex composed of cellular proteins, including Cullin5 (CUL5), Elongin B (EloB), and Elongin C (EloC) (Vif–CUL5–EloB/C complex) [64,68,69,70,71,72]. This complex mediates the polyubiquitination of APOBEC3G, marking it for degradation via the 26S proteasome. Although this process is generally dependent on specific lysine residues in APOBEC3G, Vif has been shown to promote APOBEC3G degradation even when these lysine residues are mutated or absent, suggesting a broader targeting mechanism [64].

Beyond APOBEC3G, Vif also counteracts other members of the APOBEC3 family, including APOBEC3C through APOBEC3H, via the same E3 ligase complex. Vif’s interaction with the complex is mediated by two critical motifs: a SOCS (Suppressor of Cytokine Signaling proteins)-box-like motif that facilitates binding to EloC and via and a zinc-binding motif that engages CUL5. These structural features enable Vif to effectively neutralize the APOBEC3-mediated antiviral activity and thereby enhance efficient HIV-1 replication and infectivity [73].

Recent proteomic analyses have identified core-binding factor subunit beta (CBF-β) as a key host factor in Vif function. CBFβ stabilizes the Vif–CUL5–EloB/C E3 ubiquitin ligase complex by directly interacting with Vif, particularly at residues W21 and W38. This interaction is indispensable for the complex’s structural integrity and functional efficiency. In the absence of CBF-β, the complex becomes unstable, leading to defective polyubiquitination and impaired degradation of APOBEC3G, ultimately compromising Vif’s antiviral counteraction [64,74,75].

Given that CBF-β also functions as a transcriptional cofactor involved in T cell development and differentiation, its interaction with Vif raises the possibility that HIV-1 may influence host immune cell differentiation. Further investigation into this interaction may reveal novel insights into how HIV-1 manipulates the host immune system to promote viral persistence [64].

A summary of these interactions is presented in Table 3.

### 2.4. Host Proteins That Interact with Viral Vpr

Viral Protein R (Vpr), a 14 kDa nonstructural protein of HIV-1, plays a pivotal role in arresting the host cell cycle at the G2/M checkpoint and is critical for efficient viral replication. This arrest is mediated through Vpr-induced, ubiquitin/proteasome-dependent degradation of several host proteins, which in turn enhances HIV-1 gene expression. A substantial amount of Vpr is incorporated into viral particles and is released from the major capsid protein (CA) upon viral entry into the host cell. Notably, the release of Vpr coincides with the initiation of reverse transcription, indicating that its functions are exerted early in infection, prior to the integration of the viral genome into host DNA. These functions underscore Vpr’s role in modulating host cellular processes to favor HIV-1 replication [28].

The RNA-associated early-stage antiviral factor (REAF) is a host protein that inhibits the replication of HIV-1, HIV-2, and SIV. However, HIV-1 Vpr antagonizes REAF by promoting its degradation, thereby facilitating viral replication, particularly in primary macrophages. This interaction illustrates Vpr’s capacity to subvert intrinsic immune defenses to enhance viral propagation [28,76].

In monocyte-derived macrophages (MDMs) and HeLa-CD4 cells infected with Vpr-expressing HIV-1, nuclear levels of REAF are markedly reduced within two hours post-infection. In contrast, infection with Vpr-deficient HIV-1 results in increased nuclear REAF levels as early as 30 min post-infection. These observations suggest that in the absence of Vpr, HIV-1 infection leads to an upregulation of REAF, which acts to restrict viral replication. Conversely, when Vpr is present, it facilitates REAF degradation, thereby promoting viral replication. This underscores the essential role of Vpr in neutralizing REAF-mediated restriction [28,77].

Vpr also promotes the degradation of ten-eleven translocation methylcytosine dioxygenase (TET2), a DNA demethylase, via the VprBP-DDB1 (DNA Damage Binding Protein1)-CUL4-ROC1 (Regulator of Cullins 1) E3 ubiquitin ligase complex. TET2 degradation enhances HIV-1 replication and significantly increases the production of IL-6. In THP-1 (Human Leukemia Monocytic Cell Line 1) monocytes, transfection with Vpr-containing HIV-1 results in reduced TET2 protein levels, an effect that is absent in cells infected with Vpr-deficient HIV-1. This process is dependent on the interaction between Vpr and its binding partner, VprBP. Overexpression of TET2 suppresses HIV-1 replication, whereas TET2 knockout leads to a substantial increase in viral propagation. In TET2-deficient cells, Vpr enhances viral replication by approximately 4- to 5-fold, highlighting a mechanism by which Vpr facilitates infection through modulation of host epigenetic regulators [28,78].

Additionally, Vpr impedes the nuclear translocation of interferon regulatory factor (IRF3) and nuclear factor kappa B (NF-κB), two key transcription factors in the innate immune response triggered by pathogen-associated molecular patterns (PAMPs). Vpr achieves this by interacting with karyopherins, which mediate nuclear import, thereby preventing IRF3 and NF-κB from entering the nucleus. This inhibition dampens the activation of innate immune signaling pathways, ultimately enhancing HIV-1 replication in macrophages. These findings further demonstrate how Vpr subverts host immunity to support viral transmission and persistence [28,79,80,81,82].

A summary of these interactions is presented in Table 4.

### 2.5. Host Proteins That Interact with Viral Env

The *env* gene of HIV-1 encodes the envelope glycoprotein (Env), which plays a central role in viral attachment to and entry into host cells. Env is initially synthesized as a 160 kD precursor protein, gp160, which undergoes proteolytic cleavage into two subunits: gp120 (N-terminal) and gp41 (C-terminal). These subunits remain non-covalently associated to form a trimeric “spike” structure on the viral surface. Gp41 comprises a cytoplasmic domain embedded in the viral membrane, a membrane-spanning region, and an extracellular domain responsible for mediating the conformational rearrangements required for membrane fusion. Gp120, entirely external to the viral membrane, displays a complex architecture characterized by five conserved regions (C1–C5) interspersed with five variable regions (V1–V5), which contribute to receptor binding and immune evasion [83].

During HIV-1 infection, Vpu protein is known to counteract the antiviral function of BST-2 as described in Section 2.1. In contrast, HIV-2 lacks Vpu but compensates by using its Env protein to antagonize BST-2. Co-immunoprecipitation studies have shown that HIV-1 Env can bind to that host restriction factor. However, this interaction does not result in its degradation. Instead, HIV-2 Env appears to neutralize BST-2 by sequestering it away from the plasma membrane. Overexpression of HIV-2 Env has been observed to redirect BST-2 to perinuclear compartments, particularly within the TGN, thereby preventing its trafficking to the cell surface and blocking its antiviral activity at the site of virion release [84].

In addition to its role in BST-2 antagonism, Env also interferes with the activity of SERINC5, a host factor that impairs HIV-1 infectivity by reducing the fusogenicity of viral particles. This mechanism is distinct from the well-characterized antagonism of SERINC5 by Nef. Notably, certain HIV-1 strains, such as AD8-1 and YU-2, exhibit intrinsic resistance to the ectopic expression of SERINC5, even though their respective Nef proteins do not suppress it. Research has identified the V1, V2, and V3 loops of Env as critical regions influencing resistance to SERINC5, with the V3 loop serving as a key determinant. Furthermore, comparative studies indicate that HIV-1 subtypes A, C, and D demonstrate significantly greater resistance to SERINC5 than subtype B, suggesting that subtype-specific polymorphisms in Env contribute to differential susceptibility [28,85,86].

Although Env can mitigate the antiviral effects of SERINC5, it does not prevent the incorporation of SERINC5 into newly formed virions. Instead, SERINC5 impairs HIV-1 infectivity by altering Env’s conformation, thereby compromising its fusogenicity function. This disruption leads to a block in viral fusion at a pre-hemifusion stage. Despite functional interactions, imaging studies using pseudotyped HIV-1 particles have not revealed colocalization of Env and SERINC5, indicating that SERINC5’s inhibition of fusion occurs without direct physical interaction on the viral surface [28,85,86].

A summary of these interactions is presented in Table 5.

### 2.6. Host Proteins That Interact with Viral Vpx in HIV-2

Within the HIV-2/SIVsm/SIVmac lineage, two accessory proteins, Vpr and Viral Protein X (Vpx), have been identified as homologs to the HIV-1 Vpr protein. Although HIV-2 Vpr retains functional similarities to its HIV-1 counterpart, including the ability to induce G2 phase cell cycle arrest, Vpx serves a distinct function and does not influence cell cycle progression. Instead, Vpx is essential for the effective infection of non-proliferating cells. Phylogenetic analyses suggest that Vpx originated from a duplication of the *vpr* gene within this lineage, which diverged from other primate lentiviruses. Despite their high sequence homology and shared evolutionary origin, Vpr and Vpx have functionally diverged, with Vpx acting as a paralog of HIV-1 Vpr [87].

A major host restriction factor targeted by lentiviruses is SAMHD1 (Sterile Alpha Motif and Histidine Aspartate domain-containing protein 1), which is predominantly expressed in non-dividing cells, including dendritic cells, monocytes, resting CD4+ T cells, and macrophages. SAMHD1 restricts HIV-1 replication by depleting intracellular deoxynucleoside triphosphate (dNTPs) pools required for reverse transcription [28,88]. This is achieved through its dNTPase activity, which hydrolyzes dNTPs into deoxynucleosides and inorganic triphosphates. In addition to this, SAMHD1 also possesses RNAse activity capable of degrading viral RNA, although the specific role of this function in HIV-1 restriction remains under investigation [89].

HIV-1 and related viruses, such as SIVcpz, the immediate precursor of HIV-1, remain susceptible to SAMHD1-mediated restriction due to the absence of Vpx. In contrast, HIV-2 and SIVsm encode Vpx, which counteracts SAMHD1 activity by binding to its C-terminal domain and recruiting the CUL4 E3 ubiquitin ligase complex. This interaction leads to the polyubiquitination and proteasomal degradation of SAMHD1, thereby relieving the restriction on reverse transcription and facilitating efficient viral replication [28,90,91].

Beyond its role in dNTP regulation, SAMHD1 also contributes to both innate and adaptive immune responses. In monocyte-derived DCs, SAMHD1 is involved in the presentation of HIV-1 antigens via MHC class I molecules, stimulating HIV-1-specific CTL responses. Vpx-mediated degradation of SAMHD1 has been shown to enhance antigen presentation, thereby amplifying CTL activation and leading to the elimination of infected DCs [35,92]. Paradoxically, SAMHD1 depletion also activates innate immune sensing pathways. Loss of SAMHD1 triggers DNA sensing via the cyclic GMP-AMP synthase (cGAS) and stimulator of interferon genes (STING) axis, resulting in the induction of IFN-I, potent antiviral cytokines. However, Vpx can suppress this immune activation by directly interacting with STING. Specifically, Vpx binds to the STING domain required for NF-κB activation, thereby inhibiting the downstream signaling cascade and promoting viral immune evasion [28,93].

The human silencing hub (HUSH) complex, comprising transcription activation suppressor (TASOR), M-phase phosphoprotein 8 (MPP8), and Periphilin, is another host factor involved in the transcriptional silencing of integrated proviruses. HIV-2 Vpx, but not HIV-1 Vpr, antagonizes this complex by recruiting the DCAF1 (DDB1–CUL4-associated factor 1) ubiquitin ligase adaptor protein, facilitating HUSH complex degradation, and lifting transcriptional repression of the proviral genome [28,94]. This antagonistic mechanism appears to be specific to HIV-2 and a subset of SIVs.

Overall, the functional divergence of accessory proteins such as Vpx and the differential capacity to counteract host restriction factors, including SAMHD1 and the HUSH complex, underscore key evolutionary distinctions between HIV-1 and HIV-2. These differences exemplify the intricate interplay between viral and host factors that shape viral replication efficiency and immune escape mechanisms.

A summary of these interactions is presented in Table 6.

All the interactions described in Section 2.1, Section 2.2, Section 2.3, Section 2.4, Section 2.5 and Section 2.6 are presented in Figure 2 as a schematic representation.

## 3. Advances in Host-Targeted Therapeutic Strategies for HIV Infection and Persistence

HIV exploits host cellular machinery to facilitate its replication, persistence, and evasion of immune detection. Investigating the complex interactions between HIV and host cells has led to the identification of key host factors essential to the viral life cycle. These insights have informed the development of therapeutic strategies aimed at disrupting HIV-host interactions, with the goals of inhibiting viral replication, reversing latency, and enhancing immune responses. A host-targeted therapeutic approach offers promising advantages, including the potential to limit the emergence of drug-resistant viral strains and to address the challenge of latent viral reservoirs, which remain a major obstacle to achieving a cure.

A major focus of current research is the interruption of the early stages of the viral life cycle, particularly viral entry and genome integration. HIV initiates infection by binding to the CD4 receptor and the chemokine co-receptors CCR5 or CXCR4 on the surface of host immune cells. Inhibiting these interactions has proven to be a key therapeutic strategy. CCR5 antagonists, such as maraviroc, block the virus’s ability to engage the CCR5 co-receptor, thereby preventing cell entry. Gene-editing techniques, including CRISPR-Cas9 (Clustered Regularly Interspaced Short Palindromic Repeats/CRISPR-associated protein 9) and zinc-finger nucleases, have been employed to delete the CCR5 gene in T cells, rendering them resistant to infection. The naturally occurring CCR5 mutation CCR5Δ32, which confers resistance to CCR5-tropic HIV strains, further supports the therapeutic relevance of CCR5-targeted interventions. Similarly, CXCR4 antagonists like Plerixafor have demonstrated potential in blocking the entry of CXCR4-tropic strains, and the combination of CCR5 and CXCR4 inhibitors could offer broader antiviral protection [95,96,97].

Once HIV enters the cell, it must integrate its genome into the host DNA to establish infection. This process depends on several host factors, making them attractive targets for therapeutic intervention. For example, the host protein LEDGF (Lens Epithelium-Derived Growth Factor)/p75 facilitates the tethering of the viral integrase to chromatin, enabling integration. Small molecules known as LEDGINs have been developed to inhibit this interaction [98]. Other targets include nuclear import factors such as importin-7 and NUP153 (Nucleoporin153), which are essential for transporting the viral pre-integration complex into the host cell nucleus. Inhibiting these nuclear import pathways could significantly impair HIV replication and potentially reduce the development of resistance, as the targeted processes are host-dependent rather than virus-encoded.

To date, three FDA-approved drugs specifically target HIV-host interactions to inhibit viral entry or integration. While these agents are effective in preventing new infections, their utility in treating established infections remains limited. They do not eliminate infected cells or eradicate viral reservoirs, which continue to pose a major challenge in HIV cure research.

HIV has developed sophisticated mechanisms to subvert host cellular defenses that would otherwise inhibit its replication. One prominent example is the host enzyme APOBEC3G, which introduces hypermutations into the viral genome, thereby impeding its replication. HIV counteracts the viral activity through the Vif protein, which promotes APOBEC3G degradation (see Section 2.3). Therapeutic strategies aimed at inhibiting the Vif-APOBEC3G interaction may restore APOBEC3G antiviral activity, leading to deleterious mutations in the viral genome and reduced infectivity. Similarly, SAMHD1, a cellular protein that restricts HIV replication by hydrolyzing intracellular dNTPs required for reverse transcription, is neutralized by HIV-2 and SIV protein Vpx (see Section 2.6). Pharmacological enhancement of SAMHD1 activity or blockage of Vpx function represents a promising avenue for restricting HIV replication in non-dividing cells. As mentioned, the HIV-1 Vpu protein antagonizes BST-2 (see Section 2.1). Interfering with the Vpu-BST-2 interaction could trap virions within infected cells, reducing viral dissemination and contributing to viral control.

In addition to counteracting intrinsic antiviral proteins, HIV also evades adaptive immune responses by modulating antigen presentation. The Nef protein downregulates MHC class I molecules from the surface of infected cells, thereby diminishing CTL recognition and clearance of infected targets (see Section 2.2). Therapeutic interventions that inhibit Nef function could restore MHC class I surface expression, enhance CTL-mediated clearance, and potentially improve outcomes in combination with other immunotherapies.

Despite these advances, significant challenges persist. HIV’s high mutation rate and the presence of long-lived latent reservoirs complicate efforts to achieve a definitive cure. However, targeting host dependency factors rather than viral components offers a compelling strategy. This approach not only reduces the emergence of viral resistance but also addresses the underlying mechanisms of immune evasion and persistence.

Recent efforts have focused on disrupting key HIV-host protein interactions to enhance immune recognition and restrict viral propagation. Strategies such as blocking the Vif-APOBEC3G interaction or targeting Nef to restore antigen presentation may strengthen the immune system’s capacity to identify and eliminate infected cells. Moreover, small molecules or peptides that interfere with the activity of Vpu or Nef are under investigation for their potential to restore host immune defenses.

Innovative gene editing techniques, including CRISPR/Cas9, are also being explored to eliminate specific host factors that facilitate HIV immune evasion. By selectively disrupting genes essential for viral immune escape, such approaches aim to sensitize infected cells to immune clearance and contribute to long-term viral control.

Current ART remains highly effective at suppressing HIV replication and significantly reducing HIV-related morbidity and mortality. However, ART does not eradicate the virus and requires lifelong adherence, as it does not eliminate latent viral reservoirs nor fully address the sophisticated immune evasion strategies employed by HIV. While emerging therapeutic strategies, such as gene editing techniques or broadly neutralizing antibodies, have demonstrated potential, they continue to face obstacles related to long-term efficacy, high cost, and limited accessibility.

An alternative approach involves targeting the host-virus interactions that enable immune evasion and viral persistence. Therapies that enhance the host immune system’s ability to recognize and eliminate HIV-infected cells may offer advantages over current treatment modalities. By reinforcing intrinsic immune surveillance mechanisms, these strategies could induce a more durable response, potentially reducing or eliminating the need for lifelong ART. This therapeutic shift moves beyond viral suppression, aiming instead to restore functional immune control.

Targeting host factors that contribute to the maintenance of viral latency may also provide a pathway to addressing the persistent challenge of HIV reservoirs. These reservoirs, which harbor replication-competent viruses in a transcriptionally silent state, represent a major obstacle to achieving a cure. Enhancing immune function to clear latently infected cells could facilitate a transition from viral suppression to a functional cure.

In addition, therapies directed at host-virus interactions may offer improved safety profiles compared to traditional ART. Whereas ART broadly interferes with multiple stages of the viral life cycle and is associated with adverse side effects, such as hepatotoxicity, lipid metabolism disturbances, and gastrointestinal symptoms, host-targeted approaches may exert more specific effects with fewer systemic toxicities. By selectively modulating host proteins critical to immune evasion, these therapies could provide a more refined and tolerable treatment option.

Unlike ART, which primarily functions to suppress viral replication, host-directed therapies hold the potential not only to control but also eradicate infection or maintain viral suppression in the absence of ongoing treatment. Such interventions could support the development of a functional cure, thereby transforming long-term HIV management and improving the quality of life for people living with HIV.

Recent studies have explored the development of novel antiretroviral therapies based on the understanding of virus-host interactions [99]. One promising area of investigation involves the use of therapeutic antibodies, which can target specific viral epitopes and modulate host immune responses [100]. In particular, broadly neutralizing Antibodies (bNAbs) have emerged as a potential strategy for both the prevention and treatment of HIV infection [101,102]. Clinical evaluations of bNAbs have been conducted in various populations, including HIV-exposed neonates, HIV-uninfected adults, and individuals living with HIV across different stages of infection, such as primary infection, chronic viremia, ART-suppressed individuals, and those undergoing treatment interruption [103,104,105,106,107,108]. Administration of bNAbs has generally been well-tolerated, with limited reports of serious adverse effects and no serious safety concerns. In several trials, bNAbs have demonstrated the ability to reduce and maintain suppression of plasma viremia [109,110,111,112,113,114]. Beyond their antiviral activity, anti-HIV-1 bNAbs are also being assessed in the context of HIV cure strategies due to their capacity to enhance immune effector functions and influence the size and composition of the latent HIV-1 reservoir [115,116]. A current randomized, blinded, phase 1b proof-of-concept trial conducted at HIV treatment centers in the United States is evaluating a long-acting therapeutic regimen combining lenacapavir, a capsid inhibitor, with two bNAbs, teropavimab (3BNC117-LS) and zinlirvimab (10-1074-LS). Initial findings indicate that a single administration of this combination achieved sustained HIV-1 suppression for at least 26 weeks, suggesting its potential as a durable, long-acting treatment that may enhance patient adherence and long-term viral control [117].

In parallel, gene editing technologies, including effector nucleases (TALENs), Zinc-Finger Nucleases (ZFNs), and the CRISPR/Cas9 system, are being actively evaluated as part of innovative therapeutic approaches for HIV [118,119,120,121]. Although ART is effective in suppressing HIV-1 replication, it does not eradicate proviral DNA integrated into resting CD4+ T cells, which contributes to the persistence of latent viral reservoirs [122]. CRISPR/Cas9 has been tested in preclinical studies targeting several viral genomic regions, such as NF-κB binding motifs in the U3 region of the LTR and the TAR (Transactivation Response Element) sequences in the R region. These interventions have led to the inhibition of proviral transcription, suppression of viral replication, and, in some cases, excision of integrated viral DNA from host cell chromosomes [123,124,125,126,127]. CRISPR/Cas9 has also been investigated for its potential to block HIV-1 entry by editing host cell co-receptors. For instance, CCR5 disruption via ZFNs delivered by an adenoviral vector to patient-derived CD4+ T cells, followed by autologous reinfusion, resulted in durable HIV-RNA suppression in several participants [128,129,130]. Similarly, multiple studies have achieved successful CXCR4 gene disruption using CRISPR/Cas9, which conferred resistance to HIV-1 infection in both human and rhesus macaque CD4+ T cells and was associated with reduced viral p24 antigen production [131,132,133,134].

In addition to gene excision and entry blockade, CRISPR/Cas9 technology is being examined as a potential tool for reactivating latent HIV-1 viral reservoirs. Studies have shown that catalytically dead Cas9 (dCas9), when fused with transcriptional activator domains and guided by specific sgRNAs, can induce the expression of HIV-1 latent proviruses, making infected cells visible to immune clearance mechanisms [135,136,137]. This latency-reversing strategy represents another promising avenue in the broader effort to eliminate HIV reservoirs and achieve sustained viral remission.

## 4. Conclusions—Perspectives

Therapeutic strategies targeting host proteins involved in immune evasion represent a promising direction for developing more sustainable approaches to HIV treatment. By shifting the focus from virus-centered to host-centered interventions, these strategies may yield improved long-term outcomes and open the possibility for a functional cure. Such approaches could offer significant advantages over conventional ART and other virus-directed treatments in development, particularly by addressing the persistent mechanisms HIV uses to evade immune control.

Recent progress in HIV research has emphasized immune-mediated interventions aimed at addressing the challenge of the persistent HIV reservoir, which remains one of the principal obstacles to viral eradication. Although ART effectively suppresses viral replication, it fails to eliminate latently infected cells, allowing the virus to persist. Various immune-based strategies are under investigation, including enhancement of CD8+ T cell responses, the use of bNAbs, and the application of latency-reversing agents (LRAs) to reactivate latent virus, thereby exposing infected cells to immune clearance. Additional approaches, such as cytokine-based therapies and immune checkpoint inhibitors, are being explored to strengthen host immune responses against the virus reservoir. However, achieving sustained viral remission in the absence of ART remains a significant challenge.

Innate immune cells, including macrophages, NK cells, and innate lymphoid cells (ILCs), also play a pivotal role in shaping the size and distribution of the HIV reservoir. While these cells contribute to antiviral defense, HIV has evolved sophisticated mechanisms to avoid their surveillance, facilitating the persistence of infected cells. Novel strategies are emerging to harness and enhance the antiviral functions of these innate cells. For instance, IL-15 superagonists and BCL-2 (B-cell leukemia/lymphoma 2 protein) inhibitors are being tested for their potential to promote immune-mediated clearance of reservoir cells. Nonetheless, the complexity of immune interactions with the reservoir underscores the need for combined approaches that integrate multiple arms of the immune system to achieve effective and durable HIV cure outcomes.

## Figures and Tables

**Figure 1 cimb-47-00367-f001:**
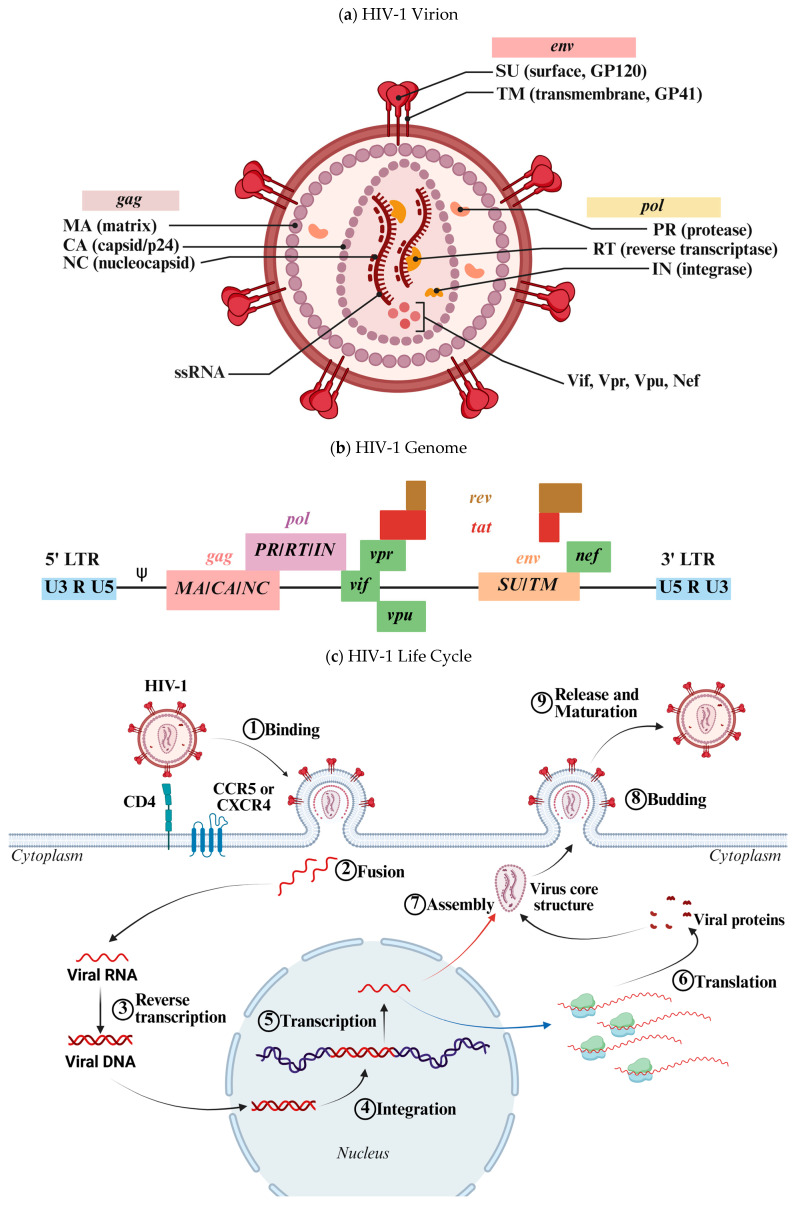
(**a**) Schematic representation of the HIV-1 virion. HIV-1 is an enveloped virus containing two molecules of negative-sense single-stranded RNA (−ssRNA). The virion includes both structural and functional proteins, as well as accessory proteins involved in viral replication and immune evasion. (**b**) Schematic representation of the HIV-1 genome. This diagram illustrates the organization of genes within the viral ssRNA. Several genes are expressed through alternative splicing mechanisms. (**c**) Schematic representation of the HIV-1 life cycle. The virus initially binds to its receptor on the host cell surface, followed by membrane fusion, which releases the viral core into the cytoplasm. There, the viral RNA is reverse-transcribed into dsDNA by the viral reverse transcriptase. The resulting DNA is transported into the nucleus and integrated into the host genome. The proviral DNA is transcribed into RNA (blue arrow), which is exported to the cytoplasm for translation. The newly synthesized viral proteins and genomic RNA (red arrow) are assembled into a new virion, which buds from the host cell membrane and undergoes maturation extracellularly. Designed by Biorender.

**Figure 2 cimb-47-00367-f002:**
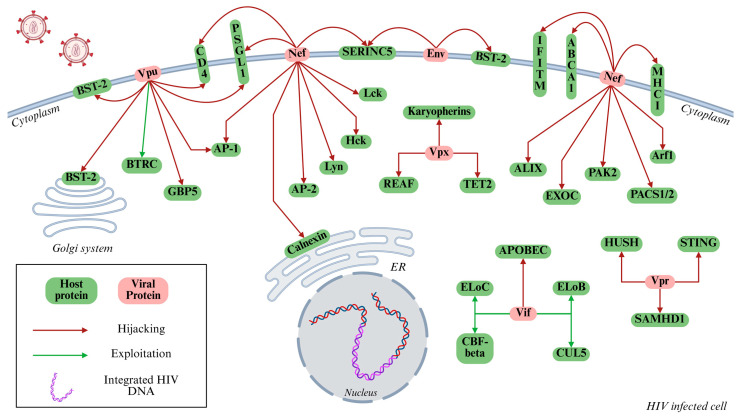
Mechanisms of HIV-mediated immune evasion in host cells. This schematic illustrates the interplay between HIV-1 encoded proteins (shown in red) and host cellular proteins (shown in green) that collectively contribute to immune evasion. Arrows indicate specific interactions through which viral proteins exploit (green arrows) or forcefully manipulate (red arrows) host cellular pathways, including immune signaling, protein degradation systems, and vesicular trafficking pathways. Key viral regulatory/accessory proteins, such as Vpu, Nef, Vif, Vpr, and Vpx, target host restriction factors, including BST-2/Tetherin, APOBEC3G, and SAMHD1, thereby suppressing innate immune responses, promoting viral persistence, and enhancing virion production. The integrated proviral DNA within the host nucleus is also depicted, signifying successful infection and exploitation of host transcriptional machinery. Designed by Biorender.

**Table 1 cimb-47-00367-t001:** Vpu-host protein interactions that result in the immune evasion of HIV-1.

Viral Protein	Host Protein	Function	Ref.
Vpu	BST-2	Inhibits the antiviral effect of BST-2	[20]
	BTRC	Acts like a SCF E3 ubiquitin-protein ligase complex for the BST-2 degradation	[20]
	AP-1	Hijacks AP-dependent trafficking pathways	[27]
	CD4	Promotes CD4 downregulation and degradation	[20]
	PSGL-1	Promotes the ubiquitination and degradation of PSGL-1	[29]
	GBP5	Counteracts GBP5 indirectly	[31]

**Table 2 cimb-47-00367-t002:** Nef-host protein interactions that result in the immune evasion of HIV-1.

Viral Protein	Host Protein	Function	Ref.
Nef	CD4	Promotes CD4 downregulation and degradation	[36]
	ALIX	Promotes lysosomal targeting of CD4 receptor	[37]
	AP-1	Hijacks AP-dependent trafficking pathways	[44]
	AP-2		[34,35]
	SERINC5	Promotes the downregulation of SERINC5 and its degradation through the endolysosomal pathway	[32]
	MHC-I	Promotes the disruption of MHC-I trafficking to the plasma membrane	[43]
	ABCA1	Promotes the dysregulation of cholesterol metabolism	[50]
	Calnexin	Promotes ABCA1 activity inhibition	[51]
	PACS-1	Association triggers MHC-I downregulation	[47]
	PACS-2		[45,47]
	ARF1		[32]
	EXOC	Promotes the inhibition of actin remodeling and interference with proximal signaling triggered by TCR engagement	[53]
	PAK2		[53]
	Lck	Influences the outcome of T-cell activation	[48]
	Hck	Promotes MCH-I downregulation	[46]
	Lyn		[46]
	PSGL-1	Promotes the downregulation of PSGL-1 and its possible redirection to intracellular compartments	[57]
	IFITM (1-3)	Alters the intracellular distribution of IFITM	[58,62]

**Table 3 cimb-47-00367-t003:** Vif-host protein interactions that result in the immune evasion of HIV-1.

Viral Protein	Host Protein	Function	Ref.
Vif	APOBEC3 family	Prevents hypermutations in viral DNA via degradation	[73]
	APOBEC3G		[66,67]
	EloB	Part of the E3 ligase complex for the degradation of APOBEC proteins	[71]
	CUL5		[70]
	EloC		[72]
	CBF-β		[75]

**Table 4 cimb-47-00367-t004:** Vpr-host protein interactions that result in the immune evasion of HIV-1.

Viral Protein	Host Protein	Function	Ref.
Vpr	REAF	Promotes the degradation of REAF, enhancing viral replication	[77]
	TET2	Promotes the degradation of TET2, enhancing viral replication	[78]
	Karyopherins	Blocks the IRF3 and NF-κB nuclear translocation	[79]

**Table 5 cimb-47-00367-t005:** Env-host protein interactions that result in the immune evasion of HIV-1 and HIV-2.

Viral Protein	Host Protein	Function	Ref.
Env	BST-2	Confines BST-2 to the trans-Golgi network (HIV-2)	[84]
	SERINC5	Unknown counteract mechanism	[85,86]

**Table 6 cimb-47-00367-t006:** Vpx-host protein interactions that result in the immune evasion of HIV-2.

Viral Protein	Host Protein	Function	Ref.
Vpx (HIV-2)	SAMHD1	Leads SAMHD1 to proteasomal degradation	[87]
	STING	Interacts with STING and inhibits the activation of NF-κB	[90]
	HUSH	Promotes the degradation of HUSH, enabling viral gene expression	[91]

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
