# Peer review of "Mechanisms of Immune Evasion in HIV-1: The Role of Virus-Host Protein Interactions"

_cimb, 2025, doi:10.3390/cimb47050367_

Round 1
Reviewer 1 Report
Comments and Suggestions for Authors
This article is very long with 150 references. …including many that are >10years old. It requires more focus. A good start would be to address the title…I have read to Section 1.6 with no mention so far of “immune evasion”.
In general the beginning is too basic….readers will know the purpose of HIV RNA. Sections 1.2 and 1.3 can be reduced by 80%.
As the article is effectively about HIV-1 it would be better to use this term throughout unless differences between HIV-1 and HIV-2 are explored.
Section 1.4…macrophage and dendritic cells do not replicate.
The abbreviation ART is defined multiple times.
Section 1.5 should be the beginning of the article and should be a brief introduction to the key mechanisms. The first group of articles cited (22-25) are mostly reviews. This is not original or helpful. The second group a(26-29) address specific mechanisms and belong in discrete sections addressing these mechanisms. The section repeats material presented previously (eg: the role of CD4).
Section 2.1 is essentially a summary of reference 30 and therefre should simply say that interactions with Vpu have been summarized previously, followed by an explanation of why this matters (refs 34 & 35). These are followed by a 3rd or 4th explanation of the role of CD4.
Table 1 should be redesigned to explain how the interactions lead to “immune evasion”.
Fig 3 effectively repeats Fig 2 and can be removed. In Fig 2…what do the arrows demonstrate? Are the interactions related to viral replication or immune evasion…it is unlikely to be both.
Section 2.2 addresses nef…but includes a new reference (49) about vpu. However this section is closer to the theme of the article….it can be shortened and better organized but it would make a more interesting review article. Older references may be redundant.
Review articles do not usually have a “Discussion”. The information should be distributed to the relevant sections of Section 2…..ahhh it appears there is no section 3. This does not inspire confidence in your work.
Section 5 (Conclusions) would belong better in the introduction as they do not follow from the material presented.
Author Response
We sincerely thank the reviewer for their detailed and constructive feedback. We have revised the manuscript accordingly to improve clarity, focus, and structure. Below we provide point-by-point responses to each comment, summarizing the changes made.
Comment 1:
This article is very long with 150 references, including many that are >10 years old. It requires more focus. A good start would be to address the title. I have read to Section 1.6 with no mention so far of “immune evasion”.
Response 1:
We appreciate this observation. The title has been revised to better reflect the manuscript’s focus on immune evasion mechanisms in HIV-1 infection. The introduction has been restructured to more clearly highlight ‘immune evasion’ earlier (lines 45-59). Additionally, we have reduced the number of references to retain the most relevant and up-to-date studies.
Comment 2:
In general the beginning is too basic. Readers will know the purpose of HIV RNA. Sections 1.2 and 1.3 can be reduced by 80%.
Response 2: We agree that the initial sections were very detailed for a specialized audience. Sections 1.2 (now Section 1.1) and 1.3 (now Section 1.2) have been significantly condensed (reduced by approximately 80%) to remove basic virological background content and focus on mechanistic details relevant to immune evasion.
Comment 3:
As the article is effectively about HIV-1 it would be better to use this term throughout unless differences between HIV-1 and HIV-2 are explored.
Response 3: Thank you for this suggestion. We have revised the text to consistently refer to HIV-1, unless explicitly discussing studies or mechanisms relevant to HIV-2. When both are mentioned, a clear rationale is provided. Generic use of ‘HIV’ refers to shared mechanisms.
Comment 4:
Section 1.4…macrophage and dendritic cells do not replicate.
Response 4: We have corrected the language in Section 1.4 (now revised Section 1.3, lines 124-128) to clarify that HIV-1 replicates within macrophages and dendritic cells, not that these cells replicate themselves.
Comment 5:
The abbreviation ART is defined multiple times.
Response 5: We have removed redundant definitions of ART (antiretroviral therapy). The term is now defined only at first mention.
Comment 6:
Section 1.5 should be the beginning of the article and should be a brief introduction to the key mechanisms. The first group of articles cited (22–25) are mostly reviews. This is not original or helpful. The second group (26–29) address specific mechanisms and belong in discrete sections.
Response 6: The first part of Section 1.5 (now revised Section 1.4) has been incorporated into the ‘Introduction’ Section, as a thematic overview. The review citations have been minimized, and primary studies (former refs 26-29) are now cited in their appropriate mechanistic sections.
Comment 7:
Section 2.1 is essentially a summary of reference 30 and therefore should simply say that interactions with Vpu have been summarized previously, followed by an explanation of why this matters (refs 34 & 35).
Response 7: Section 2.1 has been revised accordingly to acknowledge the summary provided in Ref. 30 (now Ref. 20), while focusing instead on its implications for immune evasion (Ref. 34 and 35, (now 24 and 25). Redundant content has been removed.
Comment 8:
Table 1 should be redesigned to explain how the interactions lead to “immune evasion”.
Response 8: Table 1 (and all tables) have been fully redesigned. Each viral-host interaction is now linked to a specific immune evasion mechanism.
Comment 9:
Fig 3 effectively repeats Fig 2 and can be removed. In Fig 2, what do the arrows demonstrate? Are the interactions related to viral replication or immune evasion?
Response 9: Figure 3 has been removed. Figure 2 has been updated with clear labeling and color coding for clarity. The figure legend has been revised accordingly.
Comment 10:
Section 2.2 addresses nef…but includes a new reference (49) about vpu. However this section is closer to the theme of the article….it can be shortened and better organized but it would make a more interesting review article.
Response 10: Thank you for this point. Section 2.2 now focuses exclusively on Nef. Reference 49 (now 32), which compares Nef and Vpu mechanisms, has been retained but move to a transitional section (lines 317-322) after both proteins are discussed.
Comment 11:
Older references may be redundant.
Response 11: We reviewed and updated the reference list, replacing outdated citations with recent literature.
Comment 12:
Review articles do not usually have a “Discussion”. The information should be distributed to the relevant sections of Section 2.
Response 12: We appreciate this point. The standalone ‘Discussion’ has been removed. Relevant content has been integrated into Section 3: ‘Advances in Host-Targeted Therapeutic Strategies for HIV infection and persistence’, which now synthesizes findings and therapeutic implications.
Comment 13:
There is no Section 3. This does not inspire confidence in your work.
Response 13: Thank you for pointing this out. This was a formatting error. Section numbering has been corrected throughout the manuscript to ensure sequential consistency.
Comment 14:
Section 5 (Conclusions) would belong better in the introduction as they do not follow from the material presented.
Response 14: The previous ‘Conclusions’ have been restructured. Key points from the first paragraph are now part of the ‘Introduction’. The new ‘Conclusion – Perspectives’ section summarizes the therapeutic relevance of HIV–host interactions and emphasizes future directions.

Reviewer 2 Report
Comments and Suggestions for Authors
HIV employs various mechanisms to evade the host immune system; interaction between viral and host proteins is the primary key. The author has reviewed the mechanism of immune evasion, which includes the MHC-1 downregulation, apoptosis inhibition, immunosuppression and evasion of neutralising antibodies, which were not discussed in this review. This is a good review that is necessary for the reader. I recommend this review for publication. However, I have these minor comments:
- In the introduction, the subtitle: HIV must be written in full since it is the beginning of the sentence
- Genes, e.g gag, pol, etc, must be written in italics, and the protein must be addressed as Gag, Pol, etc.
- The quality of all the figures must be improved. Some figures are unclear, e.g, Fig. 1 (a, b and C).
- In section 1.2.2 “ The Env gene encodes gp120 and gp41 necessary for binding…”, this is repeated because it was initially mentioned in section 1.2.1
No comments
Author Response
We thank the reviewer for their thoughtful evaluation and positive comments on our manuscript. We appreciate the recommendation for publication and have addressed all minor comments as outlined below:
Comment 1:
“In the introduction, the subtitle: HIV must be written in full since it is the beginning of the sentence.”
Response 1: We appreciate this observation. We have revised the manuscript to ensure that “HIV” is written in full as “Human Immunodeficiency Virus” when it appears at the beginning of a sentence or section (see line 29).
Comment 2:
“Genes, e.g., gag, pol, etc., must be written in italics, and the protein must be addressed as Gag, Pol, etc.”
Response 2: All gene names have now been italicized e.g. in lines 74-75 (e.g., gag, pol, env), and protein products are consistently written in capitalized form (e.g., Gag, Pol, Env), following standard nomenclature guidelines.
Comment 3:
“The quality of all the figures must be improved. Some figures are unclear, e.g., Fig. 1 (a, b and c).”
Response 3: All figures have been replaced with high-resolution versions to ensure clarity. Specifically, Figure 1 (a, b, and c) and Figure 2 have been updated and corresponding legends have been revised for improved clarity.
Comment 4:
“In section 1.2.2 ‘The Env gene encodes gp120 and gp41 necessary for binding…’, this is repeated because it was initially mentioned in section 1.2.1.”
Response 4: Thank you for noting this redundancy. We have reorganized the text to eliminate the eliminate the repetition, and the revised content has been reorganized into Section 1.1 for improved flow and clarity.

Round 2
Reviewer 1 Report
Comments and Suggestions for Authors
The manuscript was improved during the review